# Pathological Contribution of Extracellular Vesicles and Their MicroRNAs to Progression of Chronic Liver Disease

**DOI:** 10.3390/biology11050637

**Published:** 2022-04-21

**Authors:** Chanbin Lee, Jinsol Han, Youngmi Jung

**Affiliations:** 1Institute of Systems Biology, College of Natural Science, Pusan National University, Pusan 46241, Korea; chanbin@pusan.ac.kr; 2Department of Integrated Biological Science, College of Natural Science, Pusan National University, Pusan 46241, Korea; wlsthf1408@pusan.ac.kr; 3Department of Biological Sciences, College of Natural Science, Pusan National University, Pusan 46241, Korea

**Keywords:** extracellular vesicles, exosomes, chronic liver disease, microRNA

## Abstract

**Simple Summary:**

Extracellular vesicles (EVs) are membrane-enclosed vesicles secreted from most types of cells. EVs encapsulate many diverse bioactive cargoes, such as proteins and nucleic acid, of parental cells and delivers them to recipient cells. Upon injury, the contents altered by cellular stress are delivered into target cells and affect their physiological properties, spreading the disease microenvironment to exacerbate disease progression. Therefore, EVs are emerging as good resources for studying the pathophysiological mechanisms of diseases because they reflect the characteristics of donor cells and play a central role in intercellular communication. Chronic liver disease affects millions of people worldwide and has a high mortality rate. In chronic liver disease, the production and secretion of EVs are significantly elevated, and increased and altered cargoes are packed into EVs, enhancing inflammation, fibrosis, and angiogenesis. Herein, we review EVs released under specific chronic liver disease and explain how EVs are involved in intercellular communication to aggravate liver disease.

**Abstract:**

Extracellular vesicles (EVs) are membrane-bound endogenous nanoparticles released by the majority of cells into the extracellular space. Because EVs carry various cargo (protein, lipid, and nucleic acids), they transfer bioinformation that reflects the state of donor cells to recipient cells both in healthy and pathologic conditions, such as liver disease. Chronic liver disease (CLD) affects numerous people worldwide and has a high mortality rate. EVs released from damaged hepatic cells are involved in CLD progression by impacting intercellular communication between EV-producing and EV-receiving cells, thereby inducing a disease-favorable microenvironment. In patients with CLD, as well as in the animal models of CLD, the levels of released EVs are elevated. Furthermore, these EVs contain high levels of factors that accelerate disease progression. Therefore, it is important to understand the diverse roles of EVs and their cargoes to treat CLD. Herein, we briefly explain the biogenesis and types of EVs and summarize current findings presenting the role of EVs in the pathogenesis of CLD. As the role of microRNAs (miRNAs) within EVs in liver disease is well documented, the effects of miRNAs detected in EVs on CLD are reviewed. In addition, we discuss the therapeutic potential of EVs to treat CLD.

## 1. Introduction

The liver is damaged by several deleterious factors, such as excessive alcohol consumption, a high-calorie diet, viral infection, and genetic disorders [1,2,3]. When the liver is severely and/or chronically injured, massive hepatocyte death occurs, which is accompanied by inflammation and fibrosis [4,5]. Dying hepatocytes release various cytokines and extracellular vesicles (EVs) carrying various cargos, which affect the pathogenesis of liver diseases [6,7,8]. Among these pathological contributors, EVs play an essential role in sharing cellular information and transmitting diverse pathophysiological conditions in the liver. EVs that are nano- to micro-sized vesicles enclosed by a membrane are released from most cell types into biological fluids, such as blood, saliva, urine, and amniotic fluid [9,10]. EVs include apoptotic bodies, microvesicles, and exosomes, and they are distinguished by their biogenesis, release pathway, size, content, and function [11,12]. EVs deliver bioinformation from donor cells to adjacent or distant target cells to mediate intercellular communications. EV-carried cargoes reflect the pathophysiological state of donor cells by delivering lipids, proteins, nucleic acids, and cellular organelles of donor cells. In the liver, all types of hepatic cells produce and release EVs to assist liver homeostasis [13]. Conde-Vancells et al. [14] found that EVs and their cargoes from hepatocytes are involved in maintaining hepatic functions, such as detoxification and lipid, carbohydrate, and amino acid metabolism. In addition, Nojima et al. [15] reported that normal hepatocyte-derived EVs improved hepatocyte survival and promoted cell proliferation through intercellular communication with adjacent hepatocytes. However, injured hepatocytes increase the amount of secreted EVs loading pro-inflammatory, angiogenic, and fibrotic substances, which influences the activity of target cells, eventually promoting disease progression [16,17]. Therefore, EVs released under disease conditions can be employed as diagnostic markers for specific diseases and as treatment targets for chronic liver disease (CLD). In this review, we summarize EVs released under specific CLD conditions and discuss how EVs are involved in intercellular communication to aggravate CLD progression, focusing on miRNAs in EVs.

## 2. Extracellular Vesicles

EVs are lipid bilayer membrane-enclosed vesicles released outside the cells. EVs are classified into three subtypes: apoptotic bodies, exosomes, and microvesicles, according to their biogenesis and secretion routes into the extracellular space (Table 1 and Figure 1) [12,18]. Apoptotic bodies, the largest EVs ranging from 1–4 μm in diameter, are generated only during programmed cell death and are considered a hallmark of apoptosis [19,20,21]. In the final phase of apoptosis, dying cells are split into distinct subcellular fragments, termed apoptotic bodies or apoptosomes. These vesicles contain various residual byproducts of dying cells, such as chromatin remnants, degraded proteins, DNA fragments, and intact organelles [22]. Some apoptotic bodies deliver apoptotic contents to phagocytes, such as macrophages or dendritic cells to simulate the immune system, whereas other apoptotic bodies are quickly removed by macrophages or surrounding cells without inducing inflammation [23,24,25]. 

The exosome sizes range from 30 to 100 nm in diameter [26,27,28]. During exosome biogenesis, early endosomes are produced by inward invagination of the plasma membrane [29]. Subsequently, early endosomal membranes protrude inward to generate intraluminal vesicles (ILVs), finally forming multivesicular bodies (MVBs) [30]. Various proteins including the endosomal sorting complex required for transport (ESCRT) are involved in exosome generation [31,32,33]. ESCRT protein complexes are categorized into four types: ESCRT-0, -I, -II, and -III [34]. The ESCRT-0 complex recognizes and selects ubiquitinated proteins and recruits ESCRT-I to the endosomal membrane [35,36]. ESCRT-I gathers ESCRT-II to form the ESCRT-I/II complex, which promotes the deformation of the endosomal membrane and forms initial buds containing sorted cargoes [37,38]. ESCRT-III, which is responsible for cutting the neck of the bud, is incorporated into the complex, and forms an ILV in MVB lumen [39,40,41]. MVBs are then fused with the plasma membrane to secrete ILVs into the extracellular milieu, and the secreted ILVs are now termed exosomes [42]. Exosome biogenesis is also mediated by an ESCRT-independent pathway. Rab11 and Rab35, small GTPases, induce MVB formation and resend membrane components from endosomes to the plasma membrane [43,44,45]. Rab27a/b is involved in exosome secretion by controlling MVB docking and fusion with the plasma membrane [46]. 

Microvesicles are larger than exosomes, ranging from 100 to 1000 nm in diameter, and are known as ectosomes or microparticles [47]. Unlike exosomes, microvesicles are generated by budding and pinching outward, directly from the plasma membrane [48,49]. Although the molecular mechanism underlying microvesicle formation is less understood than that of exosome formation, recent studies have suggested that common molecular players involved in exosome formation, such as the ESCRT complex components, participate in microvesicle biogenesis. Tumor susceptibility gene 101 protein (TSG101), a subunit of ESCRT-I, interacts with arrestin domain-containing 1, causing protrusion of the plasma membrane into the extracellular space [50]. The ESCRT-III protein drives membrane curvature and fission of microvesicles [51]. Small GTPases also impact microvesicle formation by promoting actin-myosin rearrangement for microvesicle evagination. ADP-ribosylation factor 6, a member of the rho family, activates phospholipase D to recruit extracellular signal-regulated kinase (ERK) to the plasma membrane [52]. ERK phosphorylates myosin light-chain kinase, which induces contraction of the cytoskeleton, thereby stimulating microvesicle shedding [52].

Accumulating evidence has shown that microvesicles and exosomes have similar biological properties in that they encapsulate many diverse bioactive cargoes, such as cytoskeletal proteins, transporters, adhesion proteins, biogenesis-related proteins, secreted proteins, and metabolites [53,54,55]. Microvesicles and exosomes also deliver genetic material to recipient cells by transferring messenger RNA, microRNA (miRNA), mitochondrial DNA (mtDNA), and long noncoding RNA (lncRNA) [56]. Hence, they act as messengers of cell-to-cell communication by shuttling bioactive substances and genetic materials [57,58]. In addition, their cargo content differs depending on the origin and status of the parental cells [59,60]. Upon injury, contents altered by cellular stress are delivered into target cells and impact their physiological properties, spreading the disease microenvironment to exacerbate disease progression [61,62]. Therefore, microvesicles and exosomes among EVs are emerging as good resources for studying the pathophysiological mechanisms of diseases because they reflect the characteristics of donor cells and play a central role in intercellular communication. 

## 3. EVs Produced in Chronic Liver Disease

Cells release or receive EVs and use EVs as a tool for intercellular communication in both healthy and damaged livers [61]. Hence, EVs allow cells to influence the surrounding cells and alter the liver microenvironment [13]. Alterations in EVs include the amount released and their cargo composition. In damaged livers, secreted levels of EVs containing various harmful factors increase, and EVs deliver pathological cargo to target cells, thereby contributing to CLD progression [63,64]. Hence, EVs have been suggested as attractive biomarkers for diagnosing CLD and evaluating its severity. In this section, we summarize altered EVs according to the types of CLD, nonalcoholic fatty liver disease (NAFLD), alcoholic liver disease (ALD), viral hepatitis, cholestatic liver disease, and liver cancer, and briefly describe their pathogenic roles based on the results reported to date (Table 2). We mainly discuss microvesicles and exosomes because studies on apoptotic bodies in liver disease are lacking. Therefore, EVs are used as umbrella terms for exosomes and microvesicles. 

### 3.1. EVs in NAFLD

NAFLD is the most common chronic liver disease worldwide, with a prevalence of 25.2% [65]. NAFLD is a metabolic disease that occurs in people who consume little to no alcohol and includes simple steatosis, steatohepatitis (NASH), cirrhosis, and even liver cancer [66]. Lipotoxicity causes excessive hepatocyte death, which is a leading cause of NAFLD pathogenesis [67,68,69]. The damaged hepatocytes release several EVs. Srinivas et al. [70] found that the amount of circulating EVs derived from lipotoxic stressed hepatocytes was elevated in both NAFLD patients and animal models. It was also reported that lipotoxicity promoted caspase cascades triggered by the ligand-independent activation of the death 5 receptor, and activated Rho-associated coiled-coil containing protein kinase 1 (ROCK1) signaling, increasing EV secretion from hepatocytes [71]. A damage-regulated autophagy modulator (DRAM) was shown to increase EV release in patients with NAFLD by recruiting stomatin, which induces lysosomal membrane permeabilization, and DRAM deficiency reduced exosome release in HFD-fed mice [72]. Mixed lineage kinase 3 (MLK3) is also involved in increasing EV secretion from lipotoxicity-injured hepatocytes in NASH animal models [73,74]. Endoplasmic reticulum (ER) stress is a key regulator of EV release in hepatocytes exposed to lipotoxic stress. Kakazu et al. [75] demonstrated that palmitate activates inositol-requiring enzyme 1A (IRE1A), an orchestrator of ER stress, to control the de novo biogenesis of ceramide, which in turn increases EV generation in damaged hepatocytes. These findings are supported by Dasgupta et al. [76] who reported that hepatocyte-specific Ire1a-deficiency or Ire1a suppression by an inhibitor significantly reduced EV production in the livers of mice fed high fat, high fructose, and high cholesterol diets. 

EVs released from hepatocytes damaged by lipotoxicity carry various biomolecules that aggravate the pathogenesis and progression of NAFLD. MtDNA enriched in EVs is transferred into macrophages and enhances inflammation through the activation of toll-like receptor 9 (TLR9) [77]. Protein cargo within EVs also exacerbates NAFLD progression [78]. C-X-C motif ligand 10, a chemotactic factor for macrophages, in EVs induces hepatic macrophage infiltration in mice with diet-induced NASH [73]. Integrin β1 in EVs increases macrophage recruitment by facilitating monocyte adhesion to liver sinusoidal endothelial cells (LSEC) [79]. TNF-related apoptosis-inducing ligand -bearing EVs stimulated macrophage activation by inducing the production of the pro-inflammatory cytokines interleukin (IL)-1β and IL-6 [71]. Microvesicles from damaged hepatocytes delivered Vanin-1 (VNN1) to endothelial cells to promote angiogenesis, contributing to the progression of inflammation and fibrosis in NASH [80]. Povero et al. [81] also conducted protein profiling in EVs from healthy controls and NASH patients and found that proteins, such as IL-1β and intercellular cell adhesion molecule 2, retained in EVs from NASH patients were related to NASH progression. Furthermore, EVs derived from extrahepatic sources like adipocytes contribute to NAFLD progression. Adipocyte-released EVs upregulated pro-inflammatory cytokines, such as IL-6 and monocyte chemoattractant protein-1, in hepatocytes, and interfered with insulin and gluconeogenesis [82]. Obese adipose tissue-derived EVs were shuttled to HSCs and increased the expression of fibrotic markers, such as issue inhibitor matrix metalloproteinase (TIMP)-1, TIMP-4 and integrin ανβ-5, by dysregulation of TGF-β signaling [83]. Taken together, the increased production of EVs from lipotixicity-damaged hepatocytes and adipocytes is associated with NASH progression through the promotion of inflammation and fibrosis.

### 3.2. EVs in ALD

ALD is caused by excessive alcohol consumption, unlike NAFLD, and includes steatosis, alcoholic steatohepatitis, cirrhosis, and liver cancer [84]. Ethanol generates large amounts of EVs by increasing ceramide production or ESCR protein expression [85]. Moreover, alcohol obstructs autolysosomal degradation of MVB and elevates EVs release. Upregulated miR-155 by alcohol reduced the expression of multiple target genes regulating autolysosomal degradation, such as the mammalian target of rapamycin, lysosomal-associated membrane protein 1 (LAMP1), and LAMP2, and suppressed the merging of autophagosomes and MVB, enhancing EVs production in hepatocytes and macrophages [86]. Similar to NAFLD patients, patients with ALD also have higher amounts of circulating EVs in the serum than healthy people [87,88]. In alcohol-treated mice, the level of circulating EVs increased in a caspase-dependent manner, similar to the elevation of EV secretion in NAFLD [89]. These EVs are shuttled to neighboring hepatocytes or non-parenchymal cells, and contribute to ALD pathogenesis. Cytochrome P450-enriched EVs released from acetaldehyde-injured hepatocytes have been reported to trigger apoptotic signaling in undamaged target hepatocytes by increasing phospho-c-Jun N-terminal protein kinase, Bax, and active caspase-3 [90]. Alcohol-damaged hepatocytes release cluster differentiation 40 ligand (CD40L)-containing EVs, which promote macrophage activation [89]. EVs released from ethanol-damaged hepatocytes contain higher amounts of damage-associated molecular patterns (DAMP), and activate macrophages via the TLR9 pathway [91]. Ethanol-induced DRAM1 also elevated the secretion of pyruvate kinase M2 (PMK2)-enriched EVs from hepatocytes and activated macrophages, and exacerbated ALD progression [92]. Ma et al. [93] reported that apoptosis signal–regulating kinase 1/p38MAPKα signaling activated by alcohol caused oxidative stress in hepatocytes and led them to undergo apoptosis, enhancing production of mtDNA-enriched EVs from hepatocytes in an ALD mice model. It has been shown that the amount of mtDNA-enriched EVs in serum is elevated in chronically damaged liver by ethanol, promoting inflammation and hepatocyte damage [94]. Ethanol exposure also increased hepatocyte-originated exosomes having plentiful mitochondrial double strand RNA, and these exosomes enhanced IL-1β production in Kupffer cells by activating TLR3. The TLR3-dependent increase of IL-1β upregulated IL-17A in γδ T cells and CD4+ T cells, in the early and late stage of ALD, respectively [95].

In addition, crosstalk between alcohol-damaged hepatocytes and HSCs orchestrates the progression of ALD. EVs from alcohol-fed mice upregulated the expression of fibrotic markers, such as alpha-smooth muscle actin (α-SMA) and type 1 alpha 1 collagen (Col1α1) in primary HSCs from mice. Furthermore, EVs with mtDNA elevate IL-1β and IL-17 levels in hepatic macrophages in a TLR9-dependent manner, subsequently promoting the transdifferentiation of quiescent HSCs into activated HSCs [91]. Activated HSCs also released EVs to promote the activation of quiescent HSCs. EVs from activated HSCs by ethanol have a reduced level of Twist family basic helix-loop-helix transcription factor 1 (Twist1) compared to EVs from quiescent HSCs [96]. Low amounts of exosomal Twist1 inhibited expression of miR-214 targeting connective tissue growth factor (CTGF) in recipient HSCs with inactivation, promoting HSCs activation [96]. Exosomes from activated HSCs contained glycolysis-related proteins such as glucose transporter 1 and PKM2, which induced HSC activation by metabolic conversion [97].

### 3.3. EVs in Cholestatic Liver

When the liver is damaged, cholangiocytes respond to sustained pro-inflammatory stimuli and show dysregulated proliferation, commonly known as the ductular reaction [98]. Similar to hepatocytes, cholangiocytes are injured in cholestatic liver disease because of the progressive destruction of the bile ducts, the accumulation of bile acids, and the self-perpetuation of inflammation [99,100]. Activated cholangiocytes release EVs with abundant lncRNA H19, which has been shown to be correlated with the severity of cholestatic liver damage in both CLD patients and animal experimental models [101]. The exosomal lncRNA H19 interrupts bile acid homeostasis by blocking the small heterodimeric partner in hepatocytes and promotes HSC activation in the cholestatic liver [102]. It has also been reported that activated cholangiocytes secrete EVs carrying pro-inflammatory cargoes. EVs derived from cholangiocytes were rapidly taken up by Kupffer cells, and the delivered lncRNA H19 upregulated the levels of pro-inflammatory factors, including IL-6 and chemokine (C-C motif) ligand 2 in Kupffer cells [103]. Cholangiocyte-originated EVs delivered DAMP and S100 calcium binding protein A11 (S100A11) to the macrophage to activate it, promoting inflammation [104]. Although the role of EVs in cholestatic liver disease is relatively less known than that of other types of CLD, it is apparent that EVs isolated from bile are closely associated with the pathophysiology of cholangiopathies. Therefore, further studies are needed to explore the various pathogenic cargoes within EVs implicated in the pathogenesis and progression of cholestatic liver diseases.

### 3.4. Viral Hepatitis

Viral hepatitis refers to the inflammatory condition of the liver due to viral infection and is classified into five main types: hepatitis A, B, C, D, and E [105]. Among these, hepatitis B (HBV) and C (HCV) viruses cause chronic liver disease, with an estimated 296 and 130–170 million people suffering from chronic HBV and HCV, respectively, worldwide [106,107]. Similar to other CLD, EVs play an important role in multiple events in the pathogenesis of viral hepatitis. Hepatitis viruses utilize the EV formation machinery as a means of viral spread. Various viral particles use endosomal cellular complexes that are involved in the production of ILV and MVBs [108,109]. In addition, the hepatitis A virus structural protein Px was reported to increase exosome-like vesicles containing virions and viral proteins in Huh-7 cells, a human hepatoma cell line, by interacting with ALG-2-interacting protein X involved in EV biogenesis [110]. Several studies have demonstrated that virus-infected hepatocytes release large amounts of viral RNA-carrying EVs, which cause hepatic inflammation and liver fibrosis. Exosomes released from HBV-transfected hepatocytes induced natural killer group 2, member D ligand in macrophages in a myeloid differentiation factor 88-dependent manner and enhanced hepatic inflammation [111]. HBV-RNA-enriched EVs from hepatocytes upregulate type 1 interferon in dendritic cells [112]. EVs released from hepatitis virus-infected cells help the hepatitis virus to escape from immune reactions by interrupting the function of the immune cells. When EVs with HBV RNA or DNA were transferred to natural killer cells isolated from healthy people, the expression of inflammatory cytokines, such as IFN-γ and tumor necrosis factor (TNF)-α, in these cells decreased [113]. HCV-infected hepatocytes released EVs containing increased levels of galectin-9 led to the suppression of T cell activation by elevating programmed cell death-ligand-1 in monocytes [114]. In addition, EVs from hepatocytes transfected with HCV mediated polarization of monocytes into the M2 phenotype, and induced HSC activation and liver fibrosis [115]. Nonparenchymal cells infected with virus also participated in hepatitis progression. HCV particles were taken up by LSECs, and enhanced type 1 and 3 interferon in these cells [116]. These findings suggest that EVs are involved in viral expansion and the immune responses caused by viral hepatitis. 

### 3.5. EVs in Liver Cancer

Liver cancer is the fourth leading cause of cancer-related deaths worldwide [117]. Hepatocellular carcinoma (HCC) is the most common primary liver cancer, and together with cholangiocarcinoma (CCA), comprise approximately 12–15% of liver cancer cases [118]. EVs produced in the cancer microenvironment also mediate liver cancer progression by influencing cell proliferation, invasion, and metastasis. Proteome and transcriptome analyses of the contents within HCC-derived exosomes showed that they retained numerous oncogenic factors, such as caveolin, RAS related, S100A4, and S100A11, which promoted migration and invasion of immortalized hepatocytes by activating PI3K/AKT and MAPK signaling and increasing the secretion of active matrix metalloproteinases (MMP)-2 and MMP9 [119]. S100A protein family mediates cell migration, invasion, and metastasis. S100A11 protein enhanced HCC tumorigenesis and angiogenesis by inducing vascular tube formation and was found to be highly correlated with the poor survival of HCC patients [120]. In line with these findings, protein profiling of cancer-derived exosomes identified 129 proteins implicated in HCC progression. Among these, adenylyl cyclase-associated protein 1 was remarkably enriched in exosomes released from HCC cells with high metastatic potential [121]. 

In addition, both levels of 14-3-3ζ mRNA and protein in exosomes were significantly higher in HCC patients with poor survival. 14-3-3ζ expression abolished the antitumor effect of tumor-infiltrating T lymphocytes and evaded the immune response through T cell exhaustion [122]. Xie et al. [123] demonstrated that angiopoietin-2-containing exosomes were delivered into endothelial cells and induced angiogenesis by increasing AKT/eNOs and AKT/β-catenin signaling. Although lncRNAs are present in low amounts in EVs, TUC339, a 1198-base paired lncRNA, is elevated in EVs obtained from HCC cells and promotes HCC cell proliferation and adhesion [124,125]. Conigliaro et al. [126] found that CD90-positive liver cancer cells delivered the lncRNA H19 within exosomes to endothelial cells and stimulated angiogenesis. Increased exosomal SMAD family member 3 (SMAD3) in patients with HCC promotes lung metastases by enhancing circulating primary tumor adhesion [127]. HCC cells-derived exosomes transferred lysyl oxidase-like 4 between HCC cells and modulated cell-matrix adhesion and tumor invasion via activating the focal adhesion kinase/Src signaling pathway [128].

**Table 2 biology-11-00637-t002:** Summary of EVs and their cargoes associated with progression of CLD.

CLD	EVs Source	Biological Fluids	Cargo	Target	Effects	Ref.
NAFLD	Hepatocytes	plasma	mtDNA	Macrophage	Increased inflammation	[77]
	Hepatocytes	Cell culture medium, serum	CXCL10	Macrophage	Increased hepatic macrophage infiltration	[73]
	Hepatocytes	Cell culture medium, serum	Integrin β1	Monocyte	Increased adhesion to LSEC	[79]
	Hepatocytes	Cell culture medium, serum	TRAIL	Macrophage	Increased inflammation	[71]
	Hepatocytes	Cell culture medium, plasma	VNN1	LSECs	Promoted angiogenesis	[80]
	NASH patients	Serum	IL-1β, ICAM2	-	Related with NASH progression	[81]
	Adipocyte	Cell culture medium	IL-6, MCP-1	Hepatocytes	Interfered with insulin and gluconeogenesis	[82]
	Adipocyte	Cell culture medium	-	HSCs	Increased fibrotic marker expression	[83]
ALD	Hepatocytes	Cell culture medium, plasma	CYP2E1	Hepatocytes	Increased hepatocyte apoptosis	[90]
	Hepatocytes	Cell culture medium, serum	CD40L	Macrophage	Induced macrophage activation	[89]
	Hepatocytes	Cell culture medium	DAMP	Macrophage	Increased inflammation	[91]
	Hepatocytes		mtDNA	Macrophage	Enhanced HSCs activation Increased fibrotic marker expression	
	EtOH-fed mice	Serum	-	HSCs		
	Hepatocytes	Serum	PKM2	Macrophage	Increased inflammation	[92]
	Hepatocytes	Serum	mtDNA	Neutrophil Hepatocytes	Increased inflammation and hepatocyte injury	[93,94]
	Hepatocytes	Cell culture medium	mtdsRNA	Kupffer cells	Increased inflammation	[95]
	Activated HSCs	Cell culture medium	Twist1	HSCs	Enhanced HSCs activation	[96]
	Activated HSCs	Cell culture medium	GLUT1, PKM2	HSCs	Enhanced HSCs activation	[97]
Cholestatic liver	Cholangiocyte	Cell culture medium, serum	lncRNA H19	Hepatocytes HSCs	Interrupted bile homeostasis Promoted HSCs activation	[102]
	Cholangiocyte	Cell culture medium	lncRNA H19	Kupffer cells	Upregulated pro-inflammatory cytokines	[103]
	Cholangiocyte	Cell culture medium	DAMP, S100A11	Macrophage	Increased inflammation	[104]
Viral hepatitis	Hepatocytes	Cell culture medium	-	Macrophage	Increased inflammation	[111]
	Hepatocytes	Cell culture medium	HBV-RNA	Dendritic cells	Increased inflammation	[112]
	Hepatocytes	Serum	HBV-RNA/DNA	Natural Killer cells	Help hepatitis virus to escape from immune reaction	[113]
	Hepatocytes	Cell culture medium	-	Monocyte	Suppression of T-cell activation	[114]
	Hepatocytes	Cell culture medium	-	Monocyte	Enhanced HSCs activation	[115]
	LSEC	Cell culture medium	-	Hepatocytes	Controlled HCV replication	[116]
Liver cancer	HCC cells	Cell culture medium	Caveolin, RRAS, S100A4, S100A11	Immortalized hepatocytes	Stimulated migration and invasion of immortalized hepatocytes	[119]
	HCC cells	Cell culture medium	CAP1	-	Related with high metastatic potential	[121]
	HCC cells	Cell culture medium, Serum	14-3-3ζ	T-lymphocytes	Inhibited anti-tumor effects of T-lymphocytes	[122]
	HCC cells	Cell culture medium, Serum	ANGPT2	Endothelial cells	Promoted angiogenesis	[123]
	HCC cells	Cell culture medium	TUC339	HCC cells	Elevated HCC cell proliferation and adhesion.	[125]
	CD90-positive liver cancer cells	Cell culture medium	lncRNA H19	Endothelial cells	Promoted angiogenesis	[126]
	HCC cells	Cell culture medium	SMAD3	Detached HCC cells	Promotes lung metastases by enhancing circulating primary tumor adhesion	[127]
	HCC cells	Cell culture medium	LOXL4	HCC cells	Promoted invasion	[128]

CLD, chronic liver disease; NAFLD, nonalcoholic liver disease; mtDNA, mitochondrial DNA; CXCL10, C-X-C motif ligand 10; LSEC, liver sinusoidal endothelial cells; TRAIL, TNF-related apoptosis-inducing ligand; VNN1,vanin-1; IL-1β, interleukin-1β; ICAM2, intercellular cell adhesion molecule 2; ALD, alcoholic liver disease; CYP2E1, cytochrome P450 2E1; CD40L, cluster differentiation 40 ligand; DAMP, damage-associated molecular patterns; mtDNA, mitochondrial DNA; PKM2, pyruvate kinase M2; mtdsRNA, mitochondrial double strand RNA; Twist1, Twist family basic helix-loop-helix transcription factor 1; HSCs, hepatic stellate cells; GLUT1, glucose transporter 1; EtOH, ethanol; lncRNA, long noncoding RNA; S100A11, S100 calcium binding protein A11; HBV, hepatitis B virus; LSEC, liver sinusoidal endothelial cell; HCC, hepatocellular carcinoma; RRAS, Ras related; S100A4, S100 calcium binding protein A4; CAP1, adenylyl cyclase-associated protein 1; ANGPT2, angiopoietin-2; SMAD3, SMAD family member 3; LOXL4, lysyl oxidase-like 4.

## 4. Contribution of miRNAs in EVs to CLD

A growing body of evidence has revealed the contribution of various cargoes in EVs to CLD [129,130]. Among these, miRNAs are one of the most well-studied biomolecules within EVs [131]. MiRNAs are the evolutionarily conserved ncRNA of 19–24 nucleotides in length and function as post-transcriptional regulators of gene expression [132,133]. MiRNAs, transferred into recipient cells by EVs, impact cell behaviors by binding target sequences of mRNAs to interfere with the translational machinery [129,130]. In patients with CLD, elevated EVs carry multiple pathological miRNAs, resulting in the exacerbation of liver damage [134,135]. In addition, disease-specific alterations of miRNAs within EVs are in the spotlight as precise diagnostic biomarkers for distinguishing different types and progression stages of CLD [134,135]. Hence, in this section, we review the roles of EV-mediated miRNA in intercellular communications following each CLD and further explain the potential of miRNA as a disease-specific diagnostic marker (Table 3).

### 4.1. EV-Derived miRNAs in NAFLD

Along with the increase in circulating EVs released from damaged hepatocytes in patients with NAFLD and animal models mimicking NAFLD, the actions of miRNAs in EVs have recently been emphasized in the pathogenesis and progression of NAFLD [78,136]. The analysis of miRNAs in EVs isolated from patients or experimental animal models revealed a significant difference in the expression of various miRNAs. Newman et al. [136] analyzed serum EVs isolated from patients with NAFLD and found that the levels of miR-122, miR-192, and miR-128-3p were significantly higher in liver-specific asialoglycoprotein receptor 1 (ASGPR1)-positive EVs than in healthy controls, suggesting the correlation of miRNA levels with NAFLD severity. In line with these findings, Povero et al. [78] reported that miR-122 and miR-192 in liver-specific ASGPR1-positive EVs were upregulated in choline-deficient, L-amino acid-defined diet (NAFLD inducing diet) -fed mice compared to control diet-fed mice. In a rat model of NAFLD caused by high-fat and high-cholesterol diet-fed rats, miR-192-5p in EVs from injured hepatocytes was significantly increased compared to the control group [137]. MiR-192-5p-enriched EVs promoted M1 polarization by regulating the Rictor-Akt-FoxO1 axis and led to the activation of pro-inflammatory macrophages [137]. MiR-122 was also significantly higher in circulating EVs from both NAFLD patients and rodent models and inhibited AMP-activated protein kinase (AMPK) signaling [138]. Because AMPK is a major regulator of fatty acid and cholesterol metabolism, the effect of miR-122 on AMPK signaling promotes NAFLD progression by interfering with lipid metabolism [138,139]. Cholesterol exposure increased production of miR-122-5p-contained exosomes in hepatocytes, and exsomal miR-122-5p promoted hepatic inflammation by M1 polarization of the macrophage [140]. In addition, EVs from lipotoxicity-stressed hepatocytes delivered miR-128-3p to HSCs to downregulate peroxisome proliferator-activated receptor gamma and activate HSCs, promoting liver fibrosis [141]. Exosomal miR-1297 and miR-27a from damaged hepatocytes were transferred to and activated HSCs. MiR-1297-enriched exosomes activated HSCs through the phosphatase and tensin homolog (PTEN)/phosphoinositide 3-kinase/protein kinase B signaling pathway, accelerating the progression of NALFD [142]. Exosomal miR-27a downregulated PTEN-induced expression of putative protein kinase 1(PINK1) in HSCs [143]. Reduced RINK1 inhibited mitophagy inducing HSC apoptosis, and resulted in the proliferation and activation of HSCs [143,144].

### 4.2. EV-Derived miRNAs in ALD

Alcohol-damaged livers also produced significantly increased amounts of EVs containing miRNAs [88,145]. Momen-Heravi et al. [88] reported that several miRNAs, including miR-130a, miR-30a, miR-192, miR-1246, and miR-744, were significantly higher in circulating EVs from chronic EtOH-fed mice than in control mice. EVs isolated from the ALD mouse model showed the enrichment of miR-29a and miR-340, and these miRNAs were also identified in EVs from the blood of patients with ALD [145]. The increase in specific miRNAs in EVs contributes to the progression of ALD by influencing various pathways. It has been reported that alcohol elevates EV production in monocytes, a type of leukocyte, and transfers miR-27a into adjacent naive monocytes to trigger the polarization of monocytes to M2 macrophages, promoting liver fibrosis [146,147]. Alcohol causes hepatocytes to secrete miR-122-rich exosomes, which are taken up by monocytes. Delivered miR-122 blocks the heme oxygenase 1 pathway in monocytes and increases their sensitivity to lipopolysaccharide and pro-inflammatory cytokines, such as IL-1β and TNFα, contributing to the pathogenic progression of alcoholic hepatitis [148]. Additionally, exosomal miRNAs isolated from activated HSCs have been shown to induce hepatic fibrogenesis. The miR-17-92 cluster, which includes miR-17a, miR-19a, miR-19b, miR-20a, and miR-92, is known to have various functions in cellular pathways, such as inflammation, proliferation, apoptosis, and necrosis [149,150]. MiR-92 is upregulated in EVs isolated from the conditioned media of activated HSCs and the serum of alcohol-injured mice [149]. It has been reported that miR-92 regulates TGF-β by binding to the 3′ untranslated region of SMAD7, although the direct effect of EV-delivered miR-92 on HSC activation has not been studied [149,151]. To assure the impact of miR-92 on ALD to apply miR-92 for diagnostic markers, further study is required. 

### 4.3. EV-Derived miRNAs in Viral Hepatitis

Several miRNAs have been characterized in viral hepatitis, including the HBV and HCV [152]. In HBV-infected hepatocytes, miR-3 is produced and secreted extracellularly via EVs [153]. Increased miR-3 expression by HBV infection reduces suppressor of cytokine signaling (SOCS)5 expression and activates the JAK/STAT signaling pathway in macrophages to promote M1 polarization, eventually stimulating the secretion of IL-6, a pro-inflammatory cytokine that affects innate immune responses. EVs originated from HCV-infected cells also contribute to fibrosis [153]. Devhare et al. [154] showed that miR-19a contained in EVs from HCV-infected hepatocytes targeted SOCS3 to activate the STAT3-TGF-β pathway, and resulted in HSC activation. MiR-192 from an HCV-infected hepatocyte line directly influences HSC activation by upregulating TGF-β [155]. In addition, diagnostic studies of circulating miRNAs in patients have been performed. Jin et al. [156] profiled circulating miRNAs in the blood of patients with chronic hepatitis B (CHB). Among the 53 miRNAs, 22 were differentially expressed in CHB patients compared with that in healthy controls. In particular, CHB patients with high levels of HBV in the serum showed a significant increase in miR-122-5p levels compared to that in patients with low levels of HBV [156]. Meanwhile, the levels of miR-1246, miR-150-5p, miR-5787 and miR-8069 in plasma EVs were significantly lower in HBV patients than in healthy controls [157]. These findings indicate that deregulated miRNAs are involved and have potential as novel diagnostic tools for viral hepatitis. 

### 4.4. EV-Derived miRNAs in Liver Cancer

MiRNAs providing a favorable microenvironment for the proliferation of cancer cells are upregulated in EVs from HCC patients and animal models [158,159]. The levels of pro-proliferative and pro-migratory miR-222, miR-221, and miR-18a were higher, and the expression of pro-apoptotic and anti-growth miR-101, miR-122, and miR-195 was lower in the serum of patients with HCC than in those with cirrhosis [158]. Yang et al. [160] demonstrated that increased miR-3129 in plasma EVs from HCC patients promotes the proliferation of cancer cells and epithelial to mesenchymal transition by targeting thioredoxin-interacting protein, an important inhibitor of glucose uptake and cell proliferation. MiR-378b in EVs derived from HCC cells promoted angiogenesis by targeting TGF-β receptor III in endothelial cells [161]. MiRNA-584-5p in EVs from HCC cells also induced angiogenesis by inhibiting phosphoenolpyruvate carboxykinase 1-mediating nuclear factor erythroid-2-related factor 2 activation [162]. Highly metastatic HCC cells secrete EVs carrying miR-1247-3p, which directly targets β-1,4-Galactosyltransferase 3 and activates β1-integrin-NF-κB signaling in fibroblasts [163]. Activated fibroblasts secrete pro-inflammatory cytokines promoting cancer progression [163]. The MiRNA profiling of exosomes from CCA, the second most common primary malignant liver cancer with aberrant proliferation of cholangiocytes, showed that miR-205-5p was highly upregulated in CCA-derived exosomes compared to that in normal cholangiocytes, and its expression level paralleled the capacities of invasion and migration of CCA cells [164]. Ni et al. revealed that exosomal miR-23a-3p from CCA cells promoted tumor growth and metastasis by downregulating dynamin-3 [165].

In addition to their roles as influencers of liver cancer progression, exosomal miRNAs have been suggested as biomarkers for predicting the recurrence of HCC [166,167]. Exosomal miR-718, which suppresses HCC cell proliferation by targeting homeobox B8, is related to cancer prognosis. Sugimachi et al. [168] found that miR-718 expression was higher in circulating EV from HCC patients who underwent liver transplantation without recurrence, whereas its expression was lower in EVs from patients with HCC recurrence, suggesting that the level of exosomal miR-718 is related to HCC aggressiveness. A recent study analyzing clinicopathological data presents the significant upregulation of exosomal miR-92a-3p in patients with liver cancer having tumors larger than 5cm in diameter [169]. Li et al. [170] also revealed that the expression of miR-191, miR-486-3p, miR-1274b, miR-16, and miR-484 was significantly increased in circulating EVs isolated from the bile of CCA patients compared with those isolated from bile of patients with bile leak syndrome, benign biliary obstruction, or primary sclerosing cholangitis. 

Based on the disease-specific expression and stability of miRNAs in EVs and the easier detection with non-invasive techniques, they are considered diagnostic markers for specific diseases [171,172]. Hence, numerous preclinical studies on miRNAs in EVs have been conducted [129,130,173]. However, most studies have focused on profiling miRNAs in EVs from specific liver diseases using meta-analysis without providing the detailed roles and mechanisms explaining their roles. Using miRNAs in EVs as markers for the diagnosis or prognosis of disease is still in an early stage because of the lack of clinical evidence for miRNAs in EVs. Therefore, further research to elucidate their roles and relevant mechanisms in liver cancer is necessary to establish them as reliable and novel diagnostic markers for liver diseases.

**Table 3 biology-11-00637-t003:** Summary of EV-associated miRNAs in progression of CLD.

CLD	EVs Source	Biological Fluids	miRNA	Target	Effects	Ref.
NAFLD	NAFLD patients	Serum	Upregulated miR-122 miR-192 miR-128-3p	-	Related with NAFLD progression	[136]
	NAFLD mice model	Plasma	Upregulated miR-122 miR-192	-	Related with NAFLD progression	[78]
	Hepatocytes	Cell culture medium, Serum	miR-192-5p	Macrophages	Triggered M1 polarization	[137]
	Hepatocytes	Serum	miR-122	Hepatocytes	Dysregulated lipid metabolism	[138]
	Hepatocytes	Cell culture medium	miR-122-5p	Macrophages	Triggered M1 polarization	[140]
	Hepatocytes	Cell culture medium	miR-128-3p	HSCs	Enhanced HSCs activation	[141]
	Hepatocytes	Cell culture medium, serum	miR-1297 miR-27a	HSCs	Enhanced HSCs activation	[142,143]
ALD	EtOH-fed mice	Serum or Plasma	Upregulated miR-130a miR-30a miR-192 miR-1246 miR-744	-	Related with ALD progression	[88]
	Hepatocytes-isolated from EtOH-fed mice/ ALD patients	Cell culture medium/ Serum	Upregulated miR-29a miR-340	-	Related with ALD progression	[145]
	Monocytes	Cell culture medium	miR-27a	Naive monocytes	Promoted liver fibrosis	[146,147]
	Hepatocytes	Cell culture medium, Serum	miR-122	Monocytes	Increased inflammation	[148]
	Activated HSCs	Cell culture medium, Serum	miR-92	HSCs	Enhanced HSCs activation	[149]
Viral hepatitis	HBV-infected hepatocytes	Cell culture medium	miR-3	Macrophages	Increased inflammation	[153]
	HCV-infected hepatocytes	Cell culture medium	miR-19a	HSCs	Enhanced HSCs activation	[154]
	HCV-infected hepatocytes line	Cell culture medium	miR-192	HSCs	Enhanced HSCs activation	[155]
	Chronic hepatitis B patients	Serum	miR-122-5p	-	Related with level of HBV	[156]
	Chronic hepatitis B patients	plasma	Downregulated miR-1246 miR-150-5p miR-5787 miR-8069	-	Related with level of HBV	[157]
Liver cancer	HCC patients	Serum	Upregulated miR-222 miR-221 miR-18a	-	Elevated proliferation and migration of HCC cells	[158]
		Serum	Downregulated miR-101 miR-122 miR-195	-	Inhibited growth and apoptosis of HCC cells	
	HCC cells	plasma	miR-3129	HCC cells	Elevated HCC cells proliferation.	[160]
	HCC cells	Cell culture medium	miR-378b		Promoted angiogenesis	[161]
	HCC cells	Serum	miR-584-5p		Promoted angiogenesis	[162]
	HCC cells	Cell culture medium	miR-1247-3p	Fibroblasts	Increased inflammation	[163]
	CCA cells	Cell culture medium	miR-205-5p	CCA cells	Increased invasion and migration	[164]
	CCA cells	Cell culture medium	miR-23a-3p		Promoted tumor growth and metastasis	[165]
	HCC patients	Serum	Downregulated miR-718	-	Biomarker for predicting recurrence of HCC	[168]
	Liver cancer fibroblast /Liver cancer tissue	Cell culture medium	miR-92a-3p		Related with progress of HCC	[169]
	Bile from CCA patients	Bile	Upregulated miR-191 miR-486-3p miR-1274b miR-16 miR-484	-	Biomarker for CCA	[170]

CLD, chronic liver disease; miRNA, microRNA; NAFLD, nonalcoholic liver disease; HSCs, hepatic stellate cells; ALD, alcoholic liver disease; EtOH, ethanol; HBV, hepatitis B virus; HCV, hepatitis C virus; HCC, hepatocellular carcinoma; CCA, cholangiocarcinoma.

## 5. Possibility of EVs as Therapeutics for Liver Disease

As mentioned above, the amount of EVs released which carry pathological molecules was significantly increased in most CLD cases, and EVs, including their cargoes, participate in the onset and progression of CLD by affecting cell-cell communication. Hence, the inhibition of pathological EV generation and secretion and of the uptake of altered cargo by target cells has been proposed as a strategy to alleviate CLD. First, it is possible to interfere with the action of EVs in CLD by reducing EV production and/or secretion. A locked nucleic acid (LNA) is modified oligonucleotide as an anti-miRNA that has high and stable affinity for the target [174]. Momen-Heravi et al. [175] reported that LNA-miR-132 reduced EV secretion and liver fibrosis. Pharmacological inhibitors of MLK3 lowered the levels of pro-inflammatory cytokines-contained EVs from injured hepatocytes in a NASH mouse model [73]. The exosome biogenesis inhibitor GW4689 suppressed the formation of ILVs and the delivery of intracellular RNA from hepatocytes into HSCs [85]. GW4689 reduces CTGF-contained EVs from activated HSCs and inhibits the activation of quiescent HSCs [176]. GW4689 also inhibited exosome production from white adipose tissue and alleviated the liver damage caused by a high-fat diet [177]. Fasudil, an inhibitor of ROCK1, has been shown to reduce circulating EVs and attenuate liver injury, inflammation, and fibrosis in a NASH mouse model [71]. It also reduced the release of EVs containing CD40L in damaged hepatocytes and decreased macrophage activation and subsequent hepatic injury [89]. Another strategy is to interrupt the actions of pathological cargo loaded within the EVs. Hepatocytes stressed by lipotoxicity release sphingosine 1-phosphate (S1P)-abundant EVs and recruit macrophages. Suppression of sphingosine kinases (SK) 1 and 2, which play an important role in S1P enrichment, eliminates chemoattraction to macrophages [178]. Administration of the S1P antagonist FTY720 ameliorated liver damage, inflammation, and fibrosis in a NASH mouse model [179]. A TLR9 antagonist abrogated the action of mtDNA in EVs from HFD-fed mice to inhibit pro-inflammatory responses in macrophages in a murine NASH model [77]. Wang et al. [180] found that the ER stress inhibitor,4-phenybutric acid reduced exosomal miR-122 and attenuated liver injury in a mouse model of ALD. In addition, blocking the binding of EV to recipient cells restrains the effect of EVs on CLD. Wang et al. [181] demonstrated that pharmacological inhibitors blocking the link between fibronectin and integrin α5β1 reduced the binding of endothelial cell-derived exosomes to the HSC surface, and exosomal SK1 and S1P rarely promoted HSC activation and migration. Neutralization of VNN1 within EVs released from lipotoxicity-damaged hepatocytes abolished EV uptake by endothelial cells and hardly brought pro-angiogenic effects to these cells [80]. 

In addition, EVs could be used as delivery vehicles carrying small interfering RNAs or chemotherapeutic reagents. Because they are natural biocarriers derived from endogenous cells, EVs have many advantages, such as high in vivo stability and low cytotoxicity and immunogenicity [182,183,184]. Given that approximately 60% of intravenously injected EVs are delivered to the liver, a strategy using EVs as a delivery tool might be effective in treating liver disease [185]. It was shown that HEK293T cell-derived exosomes loading artificially RBP-J hairpin-decoy oligodeoxynucleotide (ODN) were transferred into hepatic macrophages and reduced Notch activation in macrophages, attenuating liver fibrosis [186]. Momen-Heravi et al. [187] reported that B cell-released EVs engineered with carrying miR-155 inhibitor suppressed LPS-induced TNF-α production in macrophages in vitro. In line with these findings, Zhang et al. [188] demonstrated that red blood cell-released EVs (RBC-EVs) having miR-155 antisense oligonucleotides (ASO) were phagocytosed by hepatic macrophage in a C1q-dependent manner, and miR-155 ASO alleviated acute liver failure by inducing the M2 polarization of macrophages. They also showed that RBC-EVs engineered to carry doxorubicin or sorafenib effectively delivered these therapeutic drugs into the liver, and inhibited orthotopic cancer growth. Ji et al. [189] encapsulated fibroblast-originated EVs with Clodronate to escape from non-specific phagocytosis by Kupffer cells, and then loaded nintedanib (NIN), an anti-fibrosis agent, in these EVs. These modified EVs effectively ameliorated liver fibrosis by inhibiting the proliferation of hepatic fibroblasts. These data suggest EVs as the safe and effective drug delivery tool for CLD.

However, there are many points that need to be considered, although the regulation of EV biology has recently emerged as a new strategy to treat CLD. If engineered EVs are unexpectedly delivered to cells other than the target cells, it could damage both the targeted and non-targeted cells. The inhibition of EV biogenesis, the uptake by target cells, and the action of EV contents might interfere with other vital processes, such as autophagy, cell differentiation, and cytoskeleton reorganization [190]. For example, GW4689 could disturb starvation-induced autophagy by inhibiting neutral sphingomyelinase 2 that facilitates autophagic flux by increasing Golgi-localized ceramide [191]. Imipramine, which inhibits EV release by suppressing ceramide generation causes many adverse effects, such as nightmares, nausea, drowsiness, low blood pressure, and dizziness [192]. Therefore, it is necessary to enhance the safety, efficacy, and targeting of EV inhibitors to apply them in treating liver diseases.

## 6. Conclusions

EVs are generated in all types of liver cells and provide bioinformation that reflects the characteristics of the parental cells [9,10]. Accumulating evidence indicates that they mediate intercellular communication between liver cells in hepatic pathobiology [6,7,8]. EVs derived from damaged cells are delivered into recipient cells and spread the pathological condition of EV-producing cells to EV-receiving cells, leading to alternation of the liver microenvironment [13]. In CLD, the production and secretion of EVs are significantly elevated, and their cargo composition is altered. Higher amounts of bioactive substances, including proteins and genetic materials, are packed into EVs to enhance inflammation, fibrosis, and angiogenesis [16,17]. MiRNAs are the most well-studied cargo groups within EVs. MiRNAs in EVs play important roles in CLD by regulating gene expression in target cells and accelerating disease progression [134,135]. Therefore, EVs and their cargo have attracted interest as potential biomarkers and important targets for the development of therapeutics for CLD [64]. However, several challenges remain to be overcome in EV applications in CLD, despite the great potential of EVs. The detailed mechanism underlying the characteristic changes in EVs in response to liver damage has not yet been fully elucidated; for example, how EV biosynthesis is increased and pathogenic cargoes are sorted and loaded in EVs is not known. In addition, studies on EV contents involved in CLD progression are limited to miRNAs, and research on other cargoes, including proteins and lncRNAs, is relatively insufficient. Further investigations to solve these challenges will open the door to understanding the role of EVs in the development and progression of CLD and provide promising diagnostic tools and therapeutic strategies for CLD.

## Figures and Tables

**Figure 1 biology-11-00637-f001:**
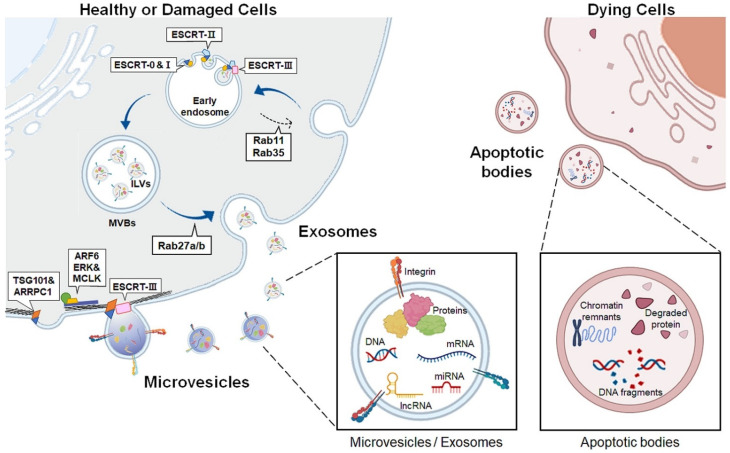
A schematic description of EVs biogenesis and secretion. The schematic representation depicts the production of broad categories of extracellular vesicles (EVs) derived from living cells (left) and apoptotic cells (right). Exosomes originate from the endosomal system. Early endosomes are generated from the inward budding of the plasma membrane, and the early endosomal membrane invaginates inward to form intraluminal vesicles (ILVs), maturing into multivesicular bodies (MVBs). In this process, the endosomal sorting complex required for transport (ESCRT) machinery participates in ILV formation. ESCRT-0 recognizes and selects ubiquitinated proteins and recruits ESCRT-I to the endosomal membrane where ESCRT-I forms ESCRT-I/II complex. ESCRT-III, cutting the neck of the bud, is added to the ESCRT-I/II complex, and forms ILVs in MVB lumen. Next, Rab27a/b mediates the docking and fusion of MVBs with plasma membranes to secrete ILVs to the extracellular space, and the released ILVs are now termed exosomes. Alternatively, early endosomes can be recycled back to the plasma membrane by Rab11 and Rab35. Microvesicles are formed by the direct budding and pinching outward from the plasma membrane. Although the molecular mechanism explaining microvesicle formation is not fully elucidated, the ESCRT machinery and small GTPases are related to their biogenesis and release. Cytoskeletal remodeling promotes microvesicles protruding through ADP-ribosylation factor 6 (ARF6)-ERK-myosin light-chain kinase (MLCK) axis. Tumor susceptibility gene 101 protein (TSG101), one of the ESCRT-1 components, and ESCRT-III proteins also mediate microvesicle release. Both exosomes and microvesicles contain various cargoes such as proteins and genetic materials, DNA, mRNA, long noncoding RNA (lncRNA), and microRNA (miRNA), reflecting the microenvironment of donor cells. On the other hand, apoptotic bodies are released by membrane blebbing from cells undergoing apoptosis. Apoptotic bodies carry a variety of cellular components such as chromatin remnants, degraded proteins and DNA fragments. This figure was created with BioRender.com.

**Table 1 biology-11-00637-t001:** Summary of characteristics of apoptotic body, exosome and microvesicle.

	Apoptotic Body	Exosome	Microvesicle
Size	1–4 μm	30–100 nm	100–1000 nm
Biogenesis	Programmed cell death	Inward protrusion of early endosomal membrane	Budding and pinching outward directly from the plasma membrane
Contents	Chromatin remnants, Degraded proteins, DNA fragments, Intact organelles	Proteins, Metabolites, DNA, mRNA, miRNA, mtDNA, lncRNA	
Biological function	Simulating the immune system	Shuttling bioactive substances and genetic materials Reflecting characteristics of donor cells Acting as messengers of cell-to-cell communication	

DNA, deoxyribonucleic acid; mRNA, messenger RNA; miRNA, microRNA; mtDNA, mitochondrial DNA; lncRNA, long noncoding RNA.

## Data Availability

Not applicable.

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
