# Peer review of "Pathological Contribution of Extracellular Vesicles and Their MicroRNAs to Progression of Chronic Liver Disease"

_biology, 2022, doi:10.3390/biology11050637_

Round 1

Reviewer 1 Report

The authors carry out an extensive review of the EVs summarizing the current knowledge on the role of EVs involved in chronic liver diseases. The review is easily readable and well structured. Although today several reviews can be found on the role of EVs in liver diseases, this review details the role of miRNAs as cargoes of EVs implicated in the different types of CLD, resulting in a very complete work in this regard.

In the field of liver diseases there are two main strategies involving EVs as therapeutic strategies: use EVs as a therapeutic target, and use EVs as delivery tools to treat liver diseases. The authors have detailed the two main strategies for using EVs as therapeutic target to treat CLD; this is either to block his release or to block the effect of its cargo. These studies demonstrate that inhibition of EV biogenesis, release and cargo ameliorates liver diseases.

As a minor point, perhaps the authors could dedicate a small paragraph to how to use EVs as tools for treatment therapy for CLD, as the authors do for example in these publications: Yang D, Liu J. Liver Int. 2020 PMID: 32593200; and Kostallari E. et. al. Adv Drug Deliv Rev. 2021 PMID: 34087329.

Author Response

As you requested, we explained the usage of EVs for CDL treatment by referring to several papers including your recommended articles: “In addition, EVs could be used as delivery vehicles carrying small interfering RNAs or chemotherapeutic reagents. Because they are natural biocarriers derived from endogenous cells, EVs have many advantages, such as high in vivo stability and low cytotoxicity and immunogenicity. Given that approximately 60% of intravenously injected EVs are delivered to the liver, a strategy using EVs as a deliver tool might be effective in treating liver disease. It was showed that HEK293T cell-derived exosomes loading artificially RBP-J hairpin-decoy oligodeoxynucleotide (ODN) were transferred into hepatic macrophages and reduced Notch activation in macrophages, attenuating liver fibrosis. Momen-Heravi et al reported that B cell-released EVs engineered with carrying miR-155 inhibitor suppressed LPS-induced TNF-α production in macrophages in vitro. In line with these findings, Zhang et al demonstrated that red blood cell-released EVs (RBC-EVs) having with miR-155 antisense oligonucleotide (ASO) were phagocytosed by hepatic macrophage in a C1q-dependent manner, and miR-155 ASO alleviated acute liver failure by inducing M2 polarization of macrophages. They also showed that engineered RBC-EVs to carry doxorubicin or sorafenib effectively delivered these therapeutic drugs into the liver, and inhibited orthotopic cancer growth. Ji et al. encapsulated fibroblast-originated EVs with Clodronate to escape from non-specific phagocytosis by Kupffer cells, and then loaded nintedanib (NIN), an anti-fibrosis agent, in these EVs. These modified EVs effectively ameliorated liver fibrosis by inhibiting proliferation of hepatic fibroblasts. These data suggest EVs as the safe and effective drug delivery tools for CLD.

Reviewer 2 Report

This manuscript demonstrated the role of extracellular vesicles (EVs) in a broad spectrum of liver diseases. It has been regarded that EVs play as crucial messengers of intercellular cross talk between cells or tissues. The roles of EV are getting attention to understand the pathophysiology of liver diseases, the understanding of EV-associated communications could be prospective molecular targets/pathways to discover effective therapeutic outcomes. Although huge efforts to discriminate EVs, many researchers use mixed term (exosomes, extracellular vesicles and microvesicles), there are still no proper markers to distinguish the nano-sized particles. Therefore, the nomenclature “Extracellular vesicles (EVs)” (a broad meaning of nano-sized particle) throughout the manuscript is very appropriate. The authors described many aspects of EVs in NAFLD, ALD, cholestatic liver, hepatitis, and cancer. Furthermore, the authors focused on the role of miRNA within EVs in the progression of liver diseases. The concept in this manuscript is very straight forward and easy to understand, however, some concerns must be addressed.

<Concerns>

  1. The authors categorized EV into apoptotic body, microvesicles and exosomes. It might be required to add a small table to show the characteristics (similarities/differences) of those extracellular vesicles.
  2. The authors mainly focused on the involvement of EVs in a various of liver diseases. And the role of miRNA in EVs were mainly explained. However, the title doesn't seem to convey enough what the authors are trying to say in their manuscript. It needs to be revised.
  3. The authors mainly focused on damaged hepatocyte-derived EVs. The involvement of non-hepatocyte derived EVs in the progression of liver disease has been studied in several publications (i.e. J Clin Invest 2021 Feb 1;131(3):e141513). Please add more information about non-hepatocyte derived EVs in CLD.
  4. In the section of “EVs in ALD”, EV-mediated mtdsRNA delivery might be demonstrated. Please add this paper “Hepatology. 2020 Aug;72(2):609-625. doi: 10.1002/hep.31041”.
  5. In line52, formation route à biogenesis
  6. In line 180, “bio-hazard factors” would be replaced.
  7. In line 186, 188, macroveislces? Or microvesicles? Make it clarify.
  8. In “2.Extracellular vesicles”, the author gave explanations in the order of apoptotic bodies, microvesicles, exosomes, but when explaining in detail, they are in the order of apoptotic bodies, exosomes, microvesicles. It must the same order.

Author Response

1. The authors categorized EV into apoptotic body, microvesicles and exosomes. It might be required to add a small table to show the characteristics (similarities/differences) of those extracellular vesicles.

: As you requested, we made a table summarizing their characteristics. The table helps to understand their characteristics at a glance, although we provided EV biogenesis in figure 1.

2. The authors mainly focused on the involvement of EVs in a various of liver diseases. And the role of miRNA in EVs were mainly explained. However, the title doesn't seem to convey enough what the authors are trying to say in their manuscript. It needs to be revised.

: As you pointed out, we changed the title: “Pathological Contribution of Extracellular Vesicles and Their MicroRNAs to Progression of Chronic Liver Disease”

3. The authors mainly focused on damaged hepatocyte-derived EVs. The involvement of non-hepatocyte derived EVs in the progression of liver disease has been studied in several publications (i.e. J Clin Invest 2021 Feb 1;131(3):e141513). Please add more information about non-hepatocyte derived EVs in CLD.

: As your comment, we added more information of non-hepatocytes (such as Adipocytes in NAFLD, HSCs in ALD, LSECs in virus hepatitis, and so on) derived EVs in CLD in the revised manuscript. Updated information is added appropriately according to the context in the revised manuscript. This is a list of added references; Am J Physiol Gastrointest Liver Physiol. 2015 Sep 15;309(6):G491-9/ FASEB J. 2019 Jul;33(7):8530-8542./ Gastroenterology. 2015 Feb;148(2):392-402.e13./ Mol Cancer. 2019 Jan 31;18(1):18/ Life Sci. 2021 May 15;273:119184./ Int J Biochem Cell Biol. 2020 Aug;125:105789./ Bioengineered. 2022 Mar;13(3):6208-6221/ Obesity. 2014 Oct;22(10):2216-23/ J Surg Res. 2014 Dec;192(2):268-75/ Crit Rev Eukaryot Gene Expr. 2022;32(1):49-57.

However, the article [J Clin Invest 2021 Feb 1;131(3):e141513] recommended by a reviewer showed the therapeutic effects, such as reduction of inflammation and fibrosis, of EVs released from neutrophil in the liver. The manuscript focused on the pathological actions of EVs. Hence, it seems to be not appropriate reference in the manuscript. Please generously consider our point.

4. In the section of “EVs in ALD”, EV-mediated mtdsRNA delivery might be demonstrated. Please add this paper “Hepatology. 2020 Aug;72(2):609-625. doi: 10.1002/hep.31041”.

: As you requested, we added the effect of EVs containing mtdsRNA in ALD subsequent to explanation for EVs carrying mtDNA: “Ethanol exposure also increased hepatocyte-originated exosomes having plentiful mitochondrial double strand RNA, and these exosomes enhanced IL-1β production in Kupffer cells by activating TLR3. TLR3-dependent increase of IL-1β upregulated IL-17A in γδ T cells and CD4+ T cells, in the early and late stage of ALD, respectively”.

5. In line52, formation route à biogenesis

: We changed ‘formation route’ with ‘biogenesis’ in the revised manuscript, as your comment.

6. In line 180, “bio-hazard factors” would be replaced.
: As you pointed out, we revised ‘bio-hazard’ with ‘harmful’.

7. In line 186, 188, macroveislces? Or microvesicles? Make it clarify.

: It is our mistake. We corrected them with microvesicles. Thank you.

8. In “2. Extracellular vesicles”, the author gave explanations in the order of apoptotic bodies, microvesicles, exosomes, but when explaining in detail, they are in the order of apoptotic bodies, exosomes, microvesicles. It must the same order.

: As you pointed out, we changed order: apoptotic bodies, exosomes, microvesicles.

Reviewer 3 Report

The review is well written and it discusses an interesting topic, but considering that new articles are published in the 2022 such us :

 PMID: 35142198;PMID: 35397694; PMID: 35377980; PMID: 35185900; and many others, it will be better to update the review with these new information.

What are the biological fluid in which EVs are studied for this review? Please add this information since the source of EVs could be different.

The authors in the title  reported "EVs and their cargo" but in the review they described just mi RNA, so it will be better to change the title with "EVs miRNA" or alternative the authors have to add more information about the EVs cargo such us protein, enzymes. etc.

Author Response

The review is well written and it discusses an interesting topic, but considering that new articles are published in the 2022 such us: PMID: 35142198 ;PMID: 35397694; PMID: 35377980; PMID: 35185900; and many others, it will be better to update the review with these new information.

: We appreciate your comments to improve the quality of the manuscript. We provided more updated information (EVs in NAFLD, ALD, viral hepatitis, and liver cancer/ miRNAs within EVs in NAFLD, viral hepatitis, and liver cancer) in the revised manuscript by referring to papers published in the last 3 years on pathological role of EVs in the liver. Updated information is added appropriately according to the context in the revised manuscript. These are added references: NAFLD [World J Gastroenterol. 2021 Apr 14;27(14):1419-1434/ Mol Ther Nucleic Acids. 2021 Nov 3;26:1241-1254./ Exp Cell Res. 2020 Feb 1;387(1):111738./ ALD [ JCI Insight. 2020 Jul 23; 5(14): e136496.], viral hepatitis [Clin Lab. 2022 Feb 1;68(2)./ Hepatology. 2020 Aug;72(2):609-625.] and liver cancer [Mol Cancer. 2019 Jan 31;18(1):18./ Life Sci. 2021 May 15;273:119184./ Int J Biochem Cell Biol. 2020 Aug;125:105789./ Crit Rev Eukaryot Gene Expr. 2022;32(1):49-57/ Bioengineered. 2022 Mar;13(3):6208-6221.].

What are the biological fluid in which EVs are studied for this review? Please add this information since the source of EVs could be different.

: As you pointed out, EV source is various and different. If EV source is provided in the main text, it seems to be redundant. Hence, we added EV source in the table 2 and 3 in the revised manuscript (table 1 and 2 in the previous version).

The authors in the title reported "EVs and their cargo" but in the review they described just mi RNA, so it will be better to change the title with "EVs miRNA" or alternative the authors have to add more information about the EVs cargo such us protein, enzymes. etc.

: We changed the title of this manuscript, as you requested” Pathological Contribution of Extracellular Vesicles and Their MicroRNAs to Progression of Chronic Liver Disease”.
